# Lactate Is a Strong Predictor of Poor Outcomes in Patients with Severe Traumatic Brain Injury

**DOI:** 10.3390/biomedicines12122778

**Published:** 2024-12-06

**Authors:** Bharti Sharma, Winston Jiang, Yashoda Dhole, George Agriantonis, Navin D. Bhatia, Zahra Shafaee, Kate Twelker, Jennifer Whittington

**Affiliations:** 1Department of Surgery, NYC Health and Hospitals, Elmhurst Hospital Center, New York, NY 11373, USA; jiangw3@nychhc.org (W.J.); agriantg@nychhc.org (G.A.); bhatian1@nychhc.org (N.D.B.); shafaeez1@nychhc.org (Z.S.); twelkerk1@nychhc.org (K.T.); harrisj20@nychhc.org (J.W.); 2Department of Surgery, Icahn School of Medicine at Mount Sinai Hospital, New York, NY 10029, USA; dholey@nychhc.org

**Keywords:** lactate, traumatic brain injury, critical care, abbreviated injury severity scale, trauma

## Abstract

**Background:** Lactate is a byproduct of glycolysis, often linked to oxygen deprivation. This study aimed to examine how lactate levels (LLs) affect clinical outcomes in patients with severe TBI, hypothesizing that higher LLs would correlate with worse outcomes. **Methods**: This is a level 1 single-center, retrospective study of patients with severe TBI between 1 January 2020 and 31 December 2023, inclusive. **Results:** Single-factor ANOVA indicated a significant decrease in LLs with increasing age. Linear regression models showed the same for hospital admission, Intensive Care Unit (ICU) admission LLs, and death LLs. Prognostic scores such as Injury Severity Scores (ISS) and Glasgow Coma Score (GCS) showed a strong correlation with both Hospital admission and ICU admission LLs. ANOVA indicated higher LLs with increasing ISS and increasing LLs with decreasing GCS. Linear regressions revealed a strong positive correlation between ISS and LLs. On linear regression, the LL measured at hospital admission and ICU admission was positively associated with the length of stay (LOS) in the hospital, LOS in the ICU, ventilator days, and mortality. Linear regression models showed that a decreased delta LL during ICU admission led to an increased LOS at the hospital and the ICU, as well as a higher number of days on a ventilator. **Discussion:** We discovered that high LLs were linked to higher AIS and GCS scores, longer stays in the hospital and ICU, more days requiring a ventilator, and higher mortality rates in patients with severe TBI. **Conclusions:** LLs can be considered a strong predictor of poor clinical outcomes in patients with severe TBI.

## 1. Introduction

Traumatic brain injury (TBI) encompasses a range of neurological injuries caused by external mechanical forces. This category includes conditions such as hematomas, contusions, and edema [1]. TBI is the leading cause of neurological disability and has a mortality rate of 30–50% among patients with severe cases [2]. Despite extensive studies on this range of brain injuries, it often proves challenging due to the various mechanisms causing TBIs and the multiple disease processes involved.

In patients with TBI, various factors such as biomarkers, electrolyte levels, intracranial pressures, and imaging studies are used to assess the initial severity of the injury and monitor its progression throughout treatment. These findings, along with the patient’s clinical presentation, are essential for risk stratification in these cases and can help determine whether a patient requires admission to a hospital or an intensive care unit (ICU) [3]. One particularly important measure is lactate levels (LLs), which can be easily obtained and rapidly processed during the initial evaluation of a TBI patient or at any point thereafter.

Lactate is a metabolic byproduct of the glycolysis pathway that increases in quantity when the final product of glycolysis, pyruvate, cannot be adequately shuttled into the pathways of aerobic metabolism. Without sufficient oxygen supply for these secondary pathways, pyruvate is converted to lactate by lactate dehydrogenase [4]. High lactate levels are often linked to hypoxic injuries, which can guide resuscitation and treatment decisions [5].

Among TBI patients specifically, high LLs indicate increased hypoxic brain injury. As described above, mitochondrial dysfunction or lack of available oxygen supply to enter secondary metabolic pathways results in lactate buildup. In TBI patients, the neuronal activity is often fueled by lactate, causing an increase in its production. Overall, high LLs indicate a more severe injury and can often predict worse outcomes in a patient’s care [6].

In the trauma bay, each TBI is evaluated by the Glasgow Coma Scale (GCS) to stratify and identify the need for specific diagnostic testing or treatment. This scale is divided into the three categories of eye, verbal, and motor response, with scores accumulating to a range from 3 to 15. A severe TBI is classified by a GCS score between 3 and 8. Moderate TBIs range from a GCS of 9 to 12, and mild cases have scores of 13–15 [7].

The Abbreviated Injury Scale (AIS), last updated in 2015, is an anatomically based scoring system that has been thought of as the gold standard for scoring injury severity [8]. This scoring system grades brain injury on a scale of 1 to 6, indicating a decreasing probability of survival with an increasing score [9]. This injury severity scale utilizes the following scoring system: Scores of 1 and 2 indicate minor and moderate injuries, respectively. A score of 3 signifies a severe injury that is not life-threatening. Scores of 4, 5, and 6 represent increasingly critical conditions: a potentially life-threatening injury with likely survival, a critical injury with uncertain survival, and an unsurvivable injury. The scale considers various factors, including scalp injuries, skull fractures, intracranial injuries, hypoxic or ischemic brain damage, and concussive injuries in its assessments [10]. The injury severity score (ISS) is an expanded version of the AIS, which considers the six body regions of the head, face, chest, abdomen, extremities (including pelvis), and external. Each of these regions is allocated an AIS score, all of which are combined to provide an ISS from 0 to 75 [11].

An important feature of this study is its completion in a hospital treating a uniquely diverse population. The center used here is a 545-bed level 1 trauma center and a major tertiary care provider within a borough of New York City. It serves an extremely ethnically diverse population of approximately one million people. With 130,042 emergency department visits in the previous year, it has one of the busiest emergency rooms among the five boroughs. It has a 12-bed Surgical Trauma ICU that is operated by surgical critical care, neurocritical care, and surgical departments [12].

There has been intensive research on TBIs and lactate, including the significance of increased LLs after TBI, the use of LLs in neurocritical care regarding treatment planning, and the use of LLs as a preferred fuel by the brain following injury [12,13]. However, we have yet to thoroughly explore the use of LLs in specific outcome prediction. Additionally, the use of AIS as an indicator of TBI severity is unique to this study, offering a more anatomically specific definition of our inclusion criteria. Through the following paper, we explain the significant correlations we found between LLs and multiple values in the categories of demographics, trauma mechanisms, severity scores, and outcomes. While we consider several metabolites in the treatment and care of our TBI patients, we found it crucial to understand the significance of each lab value. In this paper, as explained above, we focus on the contributions of LLs.

## 2. Method

This is a single-center, retrospective review conducted at a level 1 trauma center verified by the American College of Surgeons in Queens, New York City. We included all patients who presented with a severe traumatic brain injury between 1 January 2020 and 31 December 2023, inclusive. All patients with an Abbreviated Injury Severity (AIS) score of 3 or higher were included. We excluded patients who had a COVID-19 infection at the time of their injury, who died or were discharged within 24 h of their initial injury, and who had non-severe and minor injuries.

Patient data were requested from the National Trauma Registry of the American College of Surgeons (NTRACS) Database at our center (Elmhurst Hospital Center). Patients were identified based on the injury mechanism, cause of injury, primary mechanisms (lCD9 or lCDL0 E-Code), and the Abbreviated Injury Severity (AIS) score (head). The AIS score ranges from 1 to 6 per body region. Utilizing trauma registry data and adhering to inclusion and exclusion criteria, we identified a final patient group of 1125 individuals. The medical charts of the patients were reviewed, and all relevant information required for this study was collected.

We collected data using a data collection tool (Excel sheet or spreadsheet). We incorporated all data elements into this tool. Examples of data elements are demographics (for example: age, sex, race, ethnicity), AIS, GCS, ISS, TBI pattern, lactate levels, discharge disposition, mortality status, and others. The dataset underwent several preprocessing steps to ensure data integrity, confidentiality, and suitability for statistical analysis. Unique identifiers, including medical record numbers (MRNs), dates of birth (DOBs), and patient names, were removed to deidentify the dataset. This process was carried out to maintain patient privacy in compliance with ethical research standards.

The data were analyzed using both Excel and R Studio (Version: 2024.09.1+394). Laboratory values for LLs obtained at the time of admission, ICU admission, discharge, ICU discharge, and death were included in this analysis. These LLs were all collected within 24 h of the event, whether it was admission, discharge, or death. Levels taken at the initial presentation in the trauma bay were not used in this study, as they primarily serve for initial risk stratification and were not incorporated into treatment plans. Both admission and discharge LLs were examined using single-factor ANOVA, two-tailed t-tests, and linear regression models to identify correlations with demographic variables, mechanisms of injury, severity scores, hospital outcomes, and specific diagnoses.

The single-factor ANOVA model was used to examine potential correlations between LLs and the demographic variables of age and sex. In this analysis, LLs served as the dependent variable. Age was categorized into four groups: under 18, 18–45, 46–75, and over 75. Sex was categorized as either male or female. Both age and sex were compared to the LLs at various stages: hospital admission, ICU admission, hospital discharge, ICU discharge, and death. A linear regression model was utilized to compare age against these five LLs. Additionally, LLs were treated as a dependent variable when comparing blunt versus penetrating trauma using a two-tailed *t*-test. The mechanisms of trauma were also analyzed for LLs at the same five stages: hospital admission, ICU admission, hospital discharge, ICU discharge, and death.

LLs and severity score values were initially compared through the single-factor ANOVA model. Here, LLs at hospital admission, ICU admission, hospital discharge, ICU discharge, and death were divided into three categories (0–2, 2–4, >4) and utilized as the independent variable. These were analyzed against average ISS and GCS. The same five LLs were also compared to ISS and GCS with a linear regression model. Here, LLs were again used as the independent variable.

Both single-factor ANOVA and linear regression models were used to study the hospital outcome variables of hospital length of stay, ICU length of stay, days on ventilator, and mortality. For the ANOVA, LLs at hospital admission and ICU admission were divided into three categories (0–2, 2–4, >4) and utilized as the independent variable. For the linear regression model, all patients were initially divided into categories based on the number of injuries they had: one, two, three, or multiple. Then, within each category, the numerical range of LLs was used as the independent variable and compared to the same four hospital outcome variables.

Each patient in our study had associated ICD codes based on their diagnoses. The various ICD codes were then divided into the following six broad categories: subdural hematoma, subarachnoid hematoma, epidural hematoma, intraparenchymal hemorrhage, concussion, and others. Both concussion and other categories did not have sufficient data to analyze. Within each of the other four injury classification categories, LLs were again used as the independent variable and compared to hospital LOS, ICU LOS, days on ventilator, and mortality. Hospital admission and ICU admission LLs were used for the first three outcome variables, and hospital admission, ICU admission, and ICU discharge LLs were used for the analysis of mortality.

## 3. Results

For demographic data, both age and sex had some statistically significant correlations with LLs (Table 1). The single-factor ANOVA indicated a significant decrease in LLs with increasing age at hospital admission (*p* = 1.13E−16), ICU admission (*p* = 0.01), hospital discharge (*p* = 0.02), and death (*p* = 0.03). It did not show a significant correlation of the same at ICU discharge (*p* = 0.50). Linear regression also showed a significant decrease in LLs with increasing age at hospital admission (*p* = 2.65E−09), ICU admission (*p* = 0.01), and death (*p* = 0.002). The same correlations were not statistically significant at hospital discharge (*p* = 0.07) or ICU discharge (*p* = 0.54). The two-tailed T-test indicated a significant increase in LLs among male patients at hospital admission (*p* = 2.99E−05) and hospital discharge (*p* = 0.01). The same was not significant at ICU admission (*p* = 0.06), ICU discharge (*p* = 0.6), or death (*p* = 0.44). We have outlined the distributions of injury mechanisms for each of the six traumatic brain injury (TBI) diagnoses in Figure 1 of this paper. 

Concerning trauma mechanism, the two-tailed T-test did not show a significant correlation between LLs and blunt versus penetrating trauma (Table 1). LLs in penetrating trauma were insignificantly higher than blunt trauma at hospital admission (*p* = 0.46), ICU admission (*p* = 2.92), and death (*p* = 0.46). They were lower than blunt trauma at hospital discharge (*p* = 0.12) and ICU discharge (*p* = 0.25).

Single-factor ANOVA indicated significant correlations between LL ranges and ISS and GCS (Table 2). ISS increased significantly with increasing LLs at hospital admission (*p* = 5.22E−06), ICU admission (*p* = 1.92E−05), and hospital discharge (*p* = 4.62E−04). The same correlation was not significant at ICU discharge (*p* = 0.19) or death (*p* = 0.08). Similarly, GCS scores increased significantly with increasing LLs at hospital admission (*p* = 3.42E−08), ICU admission (*p* = 1.73E−09), and ICU discharge (*p* = 5.91E−07). The same correlation was not significant at hospital discharge (*p* = 0.44) or death (*p* = 0.17).

Among the overall patient population, outcome variables showed significant correlations with LLs. Single-factor ANOVA showed a significant increase in hospital LOS with increasing hospital admission (*p* = 0.008) and ICU admission (*p* = 1.79E−04) LLs. The same significant correlation was seen through linear regression at hospital admission (*p* = 0.003) and ICU admission (*p* = 0.02) (Figure 2). Single-factor ANOVA indicated a significant increase in ICU LOS with increased LLs at hospital admission (*p* = 0.046) and ICU admission (*p* = 3.25E−04). The same was seen through linear regression with *p*-values of 0.01 at hospital admission and 0.02 at ICU admission. Single-factor ANOVA showed a significant increase in days on a ventilator with increased LLs at hospital admission (*p* = 0.007) and ICU admission (*p* = 4.06E−04). The same was seen with linear regression, calculating *p*-values of 0.002 at hospital admission and 8.23E−05 at ICU admission. Single-factor ANOVA only showed a significantly increased likelihood of mortality with increased LLs at ICU admission (*p* = 0.008), not hospital admission (*p* = 0.07). Linear regressions indicated an increased likelihood of mortality with increased LLs at hospital admission (*p* = 8.13E−05), ICU admission (*p* = 3.93E−5), and ICU discharge (*p* = 2.37E−08) (Figure 3).

Linear regression models were also utilized to compare the change in LLs throughout admission to the above-mentioned four outcome variables. First off, the change in LLs from admission to ICU admission was used as the independent variable against LOS at the hospital and ICU, days on a ventilator, and mortality. This analysis did not show any significant relationship. Then, the change in LLs from ICU admission to ICU discharge was utilized as an independent variable against the same four outcomes. Here, we saw an increased hospital length of stay (*p* = 3.04E−04), ICU length of stay (5.72E−04), and number of days on a ventilator (*p* = 5.00E−06) with a decreased change in LLs. The change in lactate admissions over ICU stay did not significantly affect mortality (Table 3).

Analyzing the same four outcome values within groups of patients based on several diagnoses also produced some significant data points (Table 3). Among patients with only one severe TBI diagnosis, linear regressions indicated that both ICU LOS (*p* = 0.01) and days on ventilator (*p* = 0.048) were significantly increased with higher LLs at hospital admission. In the same group, the analysis showed a significantly increased likelihood of mortality with higher LLs at hospital admission (*p* = 0.018), ICU admission (*p* = 0.001), and ICU discharge (7.92E−04). Within the group of patients who had two diagnoses, hospital LOS (*p* = 2.80E−05), ICU LOS (*p* = 0.001), and days on the ventilator (*p* = 0.002) all increased significantly with higher LLs at ICU admission (*p* = 2.80E−05). Among the same patients, the likelihood of mortality was significantly higher among patients who had higher LLs at hospital admission (*p* = 0.04) and ICU discharge (*p* = 2.01E−04).

Among patients with three TBI diagnoses, linear regression indicated a significant increase in days on ventilators with higher LLs at ICU admission (*p* = 0.046) and an increased likelihood of mortality with higher LLs at ICU discharge (0.018) (Table 3). Within the group of patients with more than three diagnoses, the likelihood of mortality was significantly increased with higher LLs at hospital admission (*p* = 0.008), ICU admission (*p* = 0.046), and ICU discharge (*p* = 0.007) (Table 4).

The same linear regression analysis completed for data grouped by specific diagnosis also showed some important results. Among patients with a single subdural hematoma, ICU length of stay (*p* = 0.02) and days on the ventilator (*p* = 0.045) were both significantly increased with higher LLs at hospital admission. In this group, the likelihood of mortality was significantly higher with higher LLs at ICU admission (*p* = 0.02) and ICU discharge (*p* = 0.03). In patients with a single subarachnoid hematoma, the likelihood of mortality was significantly increased with higher LLs at ICU admission (*p* = 0.049). There were no significant findings through linear regression when comparing the same outcome values with LLs in patients with a single epidural hematoma. Among patients with a single intraparenchymal hemorrhage, hospital length of stay was significantly higher with increased LLs at hospital admission (*p* = 0.03) and ICU admission (*p* = 0.01). In the same group, ICU length of stay was also significantly longer with higher LLs at hospital admission (*p* = 0.01) and ICU admission (*p* = 0.03). Days on the ventilator were significantly increased with higher LLs at hospital admission (*p* = 3.25E−11). Additionally, in this group, the likelihood of mortality was significantly higher with increasing LLs at hospital admission (*p* = 0.04) and ICU discharge (*p* = 2.66E−04).

## 4. Discussion

Lactate is a crucial biomarker used for risk stratification and monitoring treatment in traumatic brain injuries, particularly in severe cases. The importance of elevated LLs in this patient population has been well understood and documented for a long time [2]. Many resources have already been allocated to better understanding the role of lactate in the overall treatment of TBIs. Research in this area has focused significantly on the formation and utilization of lactate as a potential energy source following a brain injury at the biochemical level [13]. Studies that focus on the use of LLs in outcome prediction either examine a different range of outcome variables among all TBI patients or categorize TBI severity using GCS scores [14,15].

Although the GCS score is a widely used scale for defining severe trauma diagnoses like TBIs, the AIS score used in our analysis offers a more specific, anatomically based approach by looking at identifiers of injury as opposed to symptomatic presentations seen secondary to a variety of traumatic injuries. As introduced in the background section above, the AIS score links specific anatomic injuries like lacerations, skull fractures, and brain parenchyma damage to survival probabilities through a coding system [16]. In doing so, the scoring system indicates the severity via direct calculation of the likelihood of mortality. Our work provides a unique and potentially more accurate analysis of outcome variables in patients with TBIs, since defining the severity of TBIs in this way is not commonly practiced. Furthermore, this approach encourages us to use the AIS score more frequently when assessing patients in the trauma bay [11].

The results we described above contain a variety of significant findings, indicating multiple important learning points from our study. Our analysis of demographic data showed an inverse correlation between age and LLs, with LLs decreasing with increasing age. Though it is not exactly clear why this is the case, the atrophy of the brain and its decreased mass at older ages may mean that there are fewer cells to undergo metabolic pathways and, thus, fewer opportunities to build up the byproducts of these pathways [17]. It also showed that LLs were higher in men at their hospital admission and hospital discharge. While the specific mechanism behind this remains unclear, the higher percentage of muscle mass in males may contribute. Furthermore, research on TBI and sex indicates that female sex hormones may have neuroprotective benefits post-TBI and that mitochondrial dysfunction may be increased in males. Unfortunately, TBI research is often contradictory on this topic, but some of these theories may have played a role here [18,19,20].

Among the many analyses described above, we would like to highlight those identifying the possible effects of LLs on severity scores and outcome variables. In our data, the analysis comparing LLs at hospital admission, ICU admission, and ICU discharge against ISS and GCS scores indicated a significant, direct correlation. Not only does this show that LLs during these specific time points in a patient’s admission can be used as predictors of the severity of injury, but it also confirms the validity of both ISS and GCS scoring systems, as both have a similar response. This is especially powerful for ISS as it shows a similar response to the more widely utilized and accepted GCS score.

Similarly, our analyses of LLs at hospital admission and ICU admission against hospital LOS, ICU LOS, days on the ventilator, and mortality were crucial to this study. Here, we saw that higher LLs at hospital and ICU admission caused longer hospital admissions, longer ICU admissions, and more days of mechanical ventilation. These findings are bolstered by the similar results found via single-factor ANOVA and linear regressions. Adjacent to these outcome variables, it is possible that we could extend our data analysis to include the use of LLs for predicting the need for mechanical ventilation. While our analysis does not allow for conclusions on this correlation yet, it is certainly a possible future step. Mortality was only found to be more likely with higher LLs at ICU admission through single-factor ANOVA. However, a correlation was seen between increased mortality and higher LLs at hospital admission, ICU admission, and ICU discharge through linear regression. While these findings could be impactful in TBI care, it is important to note that for many of these patients, especially those requiring ICU-level care, it is possible that other, confounding variables like respiratory distress, sepsis, hypovolemia, and cardiac ischemia were driving increased LLs. While all of these patients had severe TBI diagnoses, they may have developed many of these other, unrelated diagnoses [21,22].

External validation of these outcomes-based data will allow our results to make a much larger impact. Although we have not attempted this against data found at other institutions, a significant amount of research has already been conducted utilizing LLs in outcome prediction. Most of this looks at sepsis, overall critical care, cardiac surgical patients, etc.; however, it is clear that LLs have been shown to have predictive value elsewhere [23,24,25].

While the above analysis shows that LLs, especially at hospital and ICU admission, can indicate worse overall outcomes, the change in LLs over ICU admission is also very important. As mentioned above, we found that a decreased change in LLs throughout a patient’s ICU admission would indicate longer hospital admissions, ICU admissions, and ventilator courses. It is certainly important that we can use LLs taken on admission to predict hospital outcomes, but this analysis shows the importance of trending LLs throughout a patient’s course. It has long been known that higher LLs often indicate increased severity of brain injury, based on the idea that severe TBIs will have more hypoxic brain injury than other, less critical cases. However, our work through this project has allowed us to indicate the role of these LLs in predicting tangible outcomes that are often discussed with patients and their families.

It is important to note that the analysis of LLs and outcome variables was most fruitful when looking at the overall database. Dividing our patients into groups based on specific diagnoses or several diagnoses did not provide any clear indications that LLs could be more useful in predicting outcomes in any specific group of patients. Similar to the analysis of the overall population, the outcome variables of hospital LOS, ICU LOS, days on ventilator, and mortality were generally worse off with higher LLs among patients with one, two, three, or more than three diagnoses. Among each of these subsets, LLs at different time points in the hospital course seemed to have stronger effects. Patients with one diagnosis seemed to have more of a dependence on the LLs at initial hospital admission, while those with two, three, or more diagnoses had a stronger correlation with LLs at ICU admission. The variation in this could have been secondary to the small number of patients within each category, or the pattern could be because those with more injuries are more likely to have a significant course in the ICU.

Similar to these categories, data within groups dedicated to the specific diagnoses of subdural hematoma, subarachnoid hematoma, epidural hematoma, and intraparenchymal hemorrhage varied, but mostly showed worse outcomes with higher LLs. There was no clear pattern among the different diagnoses, again possibly due to the limited number of data points within each diagnostic group.

A vital aspect of our study’s importance is the setting in which our data were collected. As we highlighted earlier, the center used in this study serves an ethnically diverse and economically disadvantaged patient population. Often, this population is overlooked in research due to issues with funding and patient outreach. Not only is it important to focus our work on this population and further explore its needs, but studying a group of patients with this level of diversity confirms that our findings are not confounded by the utilization of one specific demographic.

### 4.1. Strengths

As we have highlighted above, we believe the strengths of this study lie in our unique utilization of the AIS score to define TBI severity. We hope that the link that AIS provides between anatomical injury and survivability will offer a more specific and accurate categorization of severe TBIs, as opposed to the symptom-based GCS score that is more widely used. This study also looks at a large variety of data points, which are analyzed in various combinations. The simultaneous use of single-factor ANOVA and linear regression analysis has allowed us to bolster the multiple significant findings we have described above. Additionally, the use of this analysis both in the overall population and within specific categories based on the type of diagnosis or number of diagnoses identifies that LLs are very useful in predicting outcomes, but must be utilized within the correct context.

### 4.2. Limitations

As a single-center study with 1125 patients, it is important to note that the size of our study and its confinement to one hospital center is an important limitation of our findings. While the above findings are significant, they have not been confirmed over a wide base of hospitals and patients. Although the treatment of TBI patients used in this hospital center is, by standards of care, accepted in the country, specific protocols on the timing of imaging, initial labs obtained in the trauma bay, and hospital or ICU admission processes are certainly specific to this center and may limit reproducibility in other settings. The confinement to a single center also indicates potential concerns regarding the patient population. We have studied an extremely diverse set of patients due to the location of our hospital center, likely introducing some demographic variables that can affect data analysis. In the future, it will be important to expand our data collection to other sites to increase variability in data, grow the patient population, and externally validate the findings we have gathered thus far. We certainly feel that the information we have found through this research can impact TBI treatment and outcome prediction, but it will be more impactful if these concepts can be proven in other hospital settings as well. With this expansion, we hope to include both academic and community centers with varying settings in urban, suburban, and rural areas.

In addition to issues that stem from the size and location of our study, it is important to note that other confounding variables may have impacted our findings. Firstly, LLs are affected by TBIs, but they are not specific to brain injury alone. Multiple disease processes like respiratory distress, bowel ischemia, sepsis, and diabetic ketoacidosis cause changes in LLs [26]. There are still others, many of which are especially common in ICU patients [27,28]. Because COVID-19 was the only illness in our exclusionary criteria, we are not certain of the presence of any other reasons for elevated LLs.

Due to the retrospective nature of this study, there was also limited standardization on the timing of LL draws. The data analysis above involved LLs at hospital admission, ICU admission, hospital discharge, ICU discharge, and death. These were drawn within 24 h of admission, discharge, or death. However, 24 h is a wide window of time, and there is limited understanding of the interventions that may have begun before obtaining the LL in question. Many interventions, such as mechanical ventilation, intravenous hydration, or operative treatment, could certainly have affected LLs.

These data were collected during the COVID-19 pandemic. Patients with COVID-19 infections were excluded from the study due to a limited understanding of the illness at the time and its link to respiratory distress in patients. Among other concerns, this increased likelihood of hypoxic lung injury could have led to a confounding increase in LLs [29,30]. Additionally, it is important to note that limitations in resources caused significant changes to patient care during this time. In many hospitals within epicenters of the pandemic, like Elmhurst, it was challenging to manage severely increased volumes of patients. While lab results and trauma evaluations were conducted according to protocol, it is possible that treatment modules had to change to accommodate for limited resources. We hope to expand data collection to include enough data that fall outside of the critical months of the pandemic.

Lastly, this study was additionally limited by the lack of comorbidities and specific details of patients’ hospital courses in the dataset. The study of lactate as an individual predictor of outcomes is not sufficient on its own as there are many confounding variables, particularly in critically ill patients, that contribute both to the rising lactate and worse in-hospital outcomes. Future studies will need further background information on the patient population as well as details in the hospital course, such as pressure requirements, operations, or medications, to better identify how lactate can be used as a prognostic variable.

### 4.3. Future Directions

As discussed above, this study has helped us clarify the use of LLs in predicting outcomes for patients with severe TBIs. In the future, we hope to bolster the significant findings established through our work and expand upon the existing dataset. First off, it will be crucial to continue collecting data from patients with TBIs. While we will certainly continue data collection at the initial center, we also hope to expand this to multiple centers around the country. Involving multiple centers will not only increase the amount of data available for analysis but also involve a variety of patient populations and hospital settings. This will inevitably create variations in TBI injury mechanisms and TBI risk stratification and treatment protocols. This will strengthen our current findings and make our data more applicable to the general public.

Additionally, it will be helpful to expand upon the specific points we have chosen to analyze. To decrease any possible confounding factors, we could consider standardizing the exact timings that LLs are collected upon hospital or ICU admission. To better understand the change in lactate, we may consider collecting LLs at more time points in patients’ hospital courses. This would allow us to further clarify the importance of acting upon a patient’s lactate at certain parts of a hospital course. Collecting lactate after the first 24 or 48 h, for example, may help to identify specific periods during which a significant change in an LL can impact the overall outcome more.

Due to the prevalence of ICU admissions for patients with severe TBIs, it may be beneficial to compare data between LLs during floor or step-down courses against those during ICU admissions. As we noted above, higher LLs are often associated with more severe TBI cases. Because of this, TBIs that result in long ICU courses may be impacted by LLs differently than those cases that are stable enough to spend longer courses on non-ICU units.

While these next steps will allow us to bolster the significant correlations we have found in our analysis thus far, we would also like to begin understanding the practical application of our findings in TBI care. It will be important to find ways to incorporate LLs in risk stratification scores in the trauma bay, upon admission, during ICU upgrade, and at discharge. As explained in the conclusion below, we will also need to find specific opportunities to incorporate the possible predictive value of LLs throughout hospital stays.

This study is part of a larger effort to understand the role of specific metabolites in the risk stratification, treatment, and recovery associated with severe TBI injuries. After we have expanded upon our understanding of LLs in this space, we will also look at other metabolites and important lab values. In the long run, it may also be helpful to combine the analysis of LLs with that of other important values like pH on blood gas.

## 5. Conclusions

While there is more work to be carried out, it is certainly clear through our study that LLs may have a significant effect on initial risk stratification and multiple complications throughout treatment. While much of this has been studied in the past, our study reinforces the validity of the AIS score to push the boundaries regarding risk stratification of TBI patients. For many years, we have been using whole-number scoring systems, like GCS, in the trauma bay for patients with TBIs. While this is effective in providing fast risk stratification in an emergency setting, it may be beneficial to expand on this to gain a more specific understanding of patients’ survivability. Although it is time-consuming, involving the use of AIS scores in real time, both in the trauma bay and upon admission, may help us directly utilize specific anatomical markers of injury in morbidity and mortality prediction. Additionally, adding biomarkers like LLs to formal risk stratification scores could provide a more intentional way to consider these lab values during TBI patient evaluation. While this will increase the complexity of initial patient risk stratification protocols, the creation of automatic score calculators can certainly help.

In addition to the use of LLs in risk stratification, it is possible that we could also use this biomarker to predict an increased need for hospital resources throughout treatment. This may allow for aid in billing and allocation of resources. Furthermore, after we have validated these data in other settings, we may also be able to utilize outcome predictions from LLs in goals-of-care conversations with families. The expanded scoring system suggested above could also be utilized in these discussions.

## Figures and Tables

**Figure 1 biomedicines-12-02778-f001:**
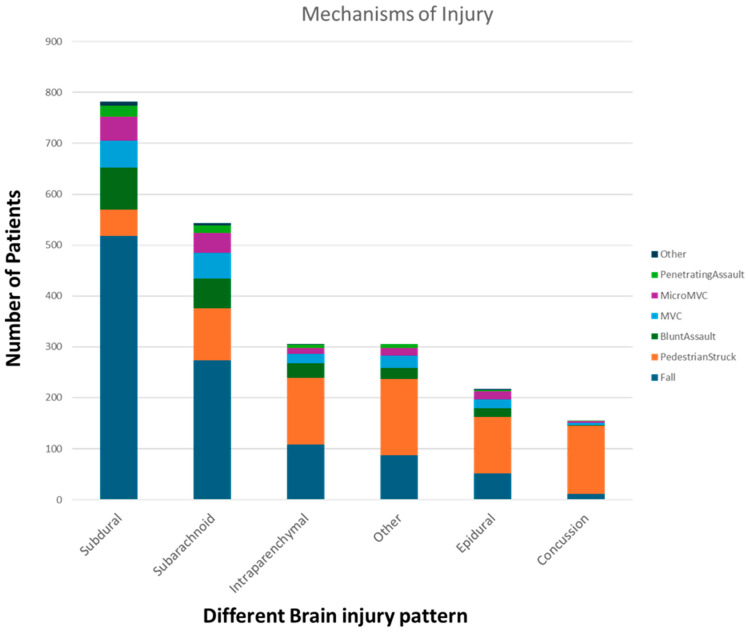
This figure outlines the distributions of injury mechanisms for each of the six traumatic brain injury (TBI) diagnoses. Each diagnosis displays the number of patients within each of the seven mechanisms of injury categories. It shows the different classifications of intracranial injuries with stratification by the mechanism of injury and the number of incidences of each that was seen in our study sample, with falls being by far the most common mechanism of injury, particularly in subdural and subarachnoid hemorrhage.

**Figure 2 biomedicines-12-02778-f002:**
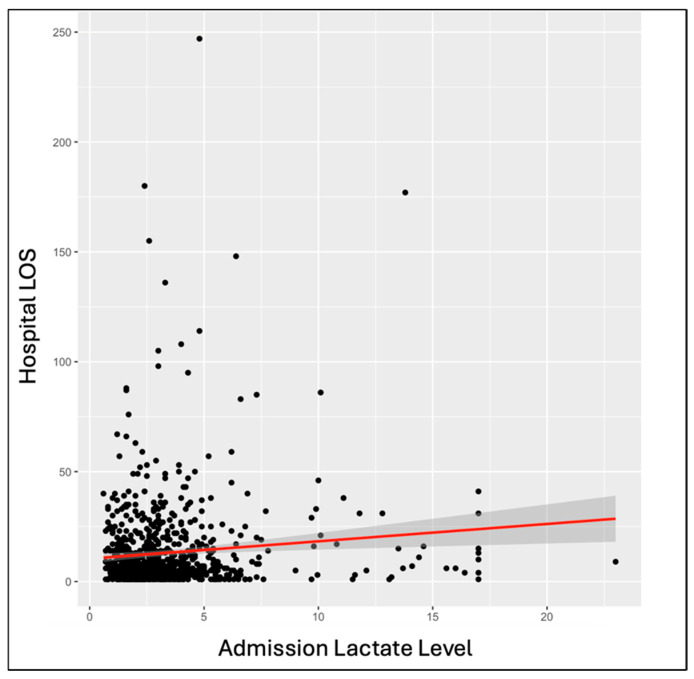
Linear regression analysis demonstrates a statistically significant correlation between admission LLs and hospital length of stay (LOS), measured in days. The line of best fit with confidence intervals shows a gradual upward trend and a positive correlation between the lactate level at admission and the hospital LOS. The correlation coefficient between lactate level at admission and hospital LOS was 0.7903.

**Figure 3 biomedicines-12-02778-f003:**
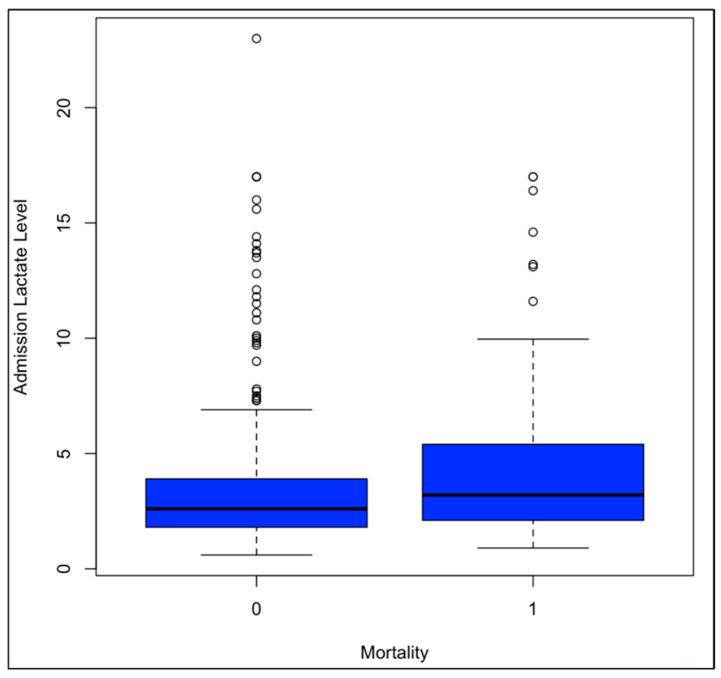
This illustrates a box plot indicating average LLs among patients who survived their injury and those who experienced death during hospitalization. Average admission LL was significantly higher for cases with mortality. Average LLs upon admission to the trauma bay among patients who survived their injury were lower than those who died during the hospitalization, with 0 indicating patients who survived and 1 indicating patients who did not.

**Table 1 biomedicines-12-02778-t001:** It represents a single-factor ANOVA analysis comparing LLs at 5 different time points (admission, ICU admission, ICU discharge, hospital discharge, death) in terms of various demographic factors. Each category shows average LLs and *p*-values. For example, under sex, one can see average LLs for female and male patients and the *p*-value for the correlation between lactate and sex for LLs at admission, ICU admission, ICU discharge, hospital discharge, and death.

		Admission	ICU Admission	ICU Discharge	Hospital Discharge	Death
Sex	Female	2.615	2.228	1.569	1.805	3.029
	Male	3.531	2.663	1.971	2.161	3.834
	*p*-value	2.999E−05	0.060	0.064	0.014	0.438
Age Range	Under 18	3.608	1.325	1.463	2.815	
	18–45	3.833	2.802	1.943	2.059	5.237
	46–74	3.308	2.606	1.977	2.207	3.123
	75+	2.402	2.121	1.656	1.812	2.918
	*p*-value	1.25E−16	0.014	0.502	0.026	0.031
Injury Type	Blunt	3.322	2.573	1.907	2.094	3.626
	Penetrating	3.824	2.922	1.178	1.342	5.3
	*p*-value	0.465	0.605	0.249	0.115	0.460
Injury Mechanism	Fall	3.346	2.438	1.880	2.135	3.328
	Blunt Assault	3.052	2.817	2.352	2.307	4.38
	MVC	3.506	2.591	1.556	1.837	2.878
	Pedestrian Struck	3.444	3.231	1.810	1.865	5.333
	Micro MVC	3.114	2.541	2.362	2.063	5.46
	Penetrating Assault	3.481	2.72	1.247	1.526	4.4
	Other	4.1	2.74	1.58	1.967	
	*p*-value	0.828	0.368	0.185	0.319	0.700
Diagnosis	Subdural	3.331	2.541	1.923	2.085	3.645
	Subarachnoid	3.526	2.604	1.874	2.061	3.426
	Epidural	3.644	2.939	1.95	1.986	2.114
	Intraparenchymal	3.544	2.792	2.015	2.111	3.807
	Concussion	2.311	1.367	1.3	1.722	1.80
	Other	3.984	2.957	1.902	1.743	3.837
	*p*-value	0.036	0.103	0.935	0.338	0.851
Number of Injuries	One	3.111	2.4	1.96	2.103	4.675
	Two	3.172	2.410	1.683	2.059	3.6
	Three	3.821	2.709	1.969	2.161	3.109
	Four+	4.213	3.133	2.114	1.65	3.917
	*p*-value	0.002	0.053	0.375	0.281	0.664

**Table 2 biomedicines-12-02778-t002:** This table shows the two-tailed t-test comparing ISS and GCS scores at different LL ranges (normal (0–2), increased (2–4), severely increased (>4)) during hospital admission, ICU admission, ICU discharge, hospital discharge, and death (wherever applicable). Each time point shows average ISS and scores for normal, increased, and severely increased LLs, along with *p*-values describing the strength of the correlation between ISS or GCS and LLs.

		Normal	Increased	Severely Increased	*p*-Value
Hospital Admission	ISS	17.426	17.798	21.234	5.226E−06
	GCS	13.722	13.011	11.602	3.423E−08
ICU Admission	ISS	19.908	20.829	25.436	1.917E−05
	GCS	12.990	11.367	9.513	1.736E−09
ICU Discharge	ISS	21.028	20.489	24.185	0.1912781
	GCS	12.535	10.736	8.667	5.907E−07
Hospital Discharge	ISS	17.995	15.685	15.824	0.0005
	GCS	13.466	13.706	13.333	0.447
Death	ISS	31.179	22.889	28.037	0.080
	GCS	9.176	8.111	6.815	0.170

**Table 3 biomedicines-12-02778-t003:** It shows the linear regression analysis comparing the change in LLs in the time between trauma bay admission and ICU admission or ICU admission and ICU discharge, showing some statistical significance. Increasing lactate at ICU admission was associated with decreased length of stay in the hospital and ICU and requiring ventilator support.

Timeframe	Outcome	*p*-Value	Coefficient
Hospital Admission to ICU Admission	Hospital LOS	0.778	−0.124
	ICU LOS	0.501	0.105
	Ventilator Days	0.322	0.148
	Mortality	0.243	0.007
ICU Admission to ICU Discharge	Hospital LOS	0.0003	−2.430
	ICU LOS	0.0006	−0.827
	Ventilator Days	5.00E−06	−1.041
	Mortality	0.204	0.012

**Table 4 biomedicines-12-02778-t004:** This table represents linear regression analysis comparing LLs at hospital admission, ICU admission, and ICU discharge with outcome variables (hospital LOS, ICU LOS, ventilator days, mortality) within categories based on some diagnosed injuries. Within each category (overall or group based on number of diagnoses), the *p*-value shows the correlation of the LL at that time point and the associated outcome variable, as well as the coefficient showing the direction of that correlation.

	Timeframe	Outcome	*p*-Value	Coefficient
Overall	Hospital Admission	Hospital LOS	0.003	0.790
		ICU LOS	0.014	0.247
		Ventilator Days	0.003	0.265
		Mortality	8.13E−05	0.016
	ICU Admission	Hospital LOS	0.016	1.357
		ICU LOS	0.023	0.457
		Ventilator Days	8.23E−05	0.753
		Mortality	3.93E−05	0.033
	ICU Discharge	Mortality	2.37E−08	0.048
One Diagnosed Injury	Admission	Hospital LOS	0.053	0.378
		ICU LOS	0.013	0.249
		Ventilator Days	0.049	0.139
		Mortality	0.018	0.010
	ICU Admission	Hospital LOS	0.668	−0.158
		ICU LOS	0.691	0.082
		Ventilator Days	0.259	0.183
		Mortality	0.001	0.029
	ICU Discharge	Mortality	0.0008	0.030
Two Diagnosed Injuries	Admission	Hospital LOS	0.285	0.627
		ICU LOS	0.753	0.067
		Ventilator Days	0.457	0.097
		Mortality	0.040	0.017
	ICU Admission	Hospital LOS	2.80E−05	5.881
		ICU LOS	0.001	1.648
		Ventilator Days	0.002	0.987
		Mortality	0.460	0.014
	ICU Discharge	Mortality	0.0002	0.069
Three Diagnosed Injuries	Admission	Hospital LOS	0.799	0.229
		ICU LOS	0.848	0.0709
		Ventilator Days	0.153	0.411
		Mortality	0.701	−0.005
	ICU Admission	Hospital LOS	0.759	0.569
		ICU LOS	0.754	0.227
		Ventilator Days	0.046	1.205
		Mortality	0.134	0.039
	ICU Discharge	Mortality	0.018	0.078
Four or More Diagnosed Injuries	Admission	Hospital LOS	0.211	1.745
		ICU LOS	0.736	−0.114
		Ventilator Days	0.759	0.190
		Mortality	0.009	0.047
	ICU Admission	Hospital LOS	0.943	0.147
		ICU LOS	0.918	−0.049
		Ventilator Days	0.759	1.318
		Mortality	0.009	0.049
	ICU Discharge	Mortality	0.007	0.079

## Data Availability

The data was requested from the Elmhurst Trauma registry and extracted using electronic medical records after receiving approval from the Institutional Review Board at our facility (Elmhurst Hospital Center).

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
