# Peer review of "Lactate Is a Strong Predictor of Poor Outcomes in Patients with Severe Traumatic Brain Injury"

_biomedicines, 2024, doi:10.3390/biomedicines12122778_

Round 1
Reviewer 1 Report
Comments and Suggestions for Authors
In this study, the authors examined the influence of lactate levels (LLs) on clinical outcomes in patients with severe traumatic brain injury (TBI). This topic aligns well with the scope of the journal, and the paper provides valuable insights. However, the manuscript would benefit from several improvements. Please see my major comments below for suggested revisions.
Major comments
1) Given the multiple statistical tests used, did the authors correct for multiple comparisons? If not, could this have affected the reported p-values, and would a correction (e.g., Bonferroni correction) change any significant findings?
2) Figures should be corrected with clear X and Y axis.
3) The manuscript suggests lactate levels (LLs) as a predictor of poor outcomes in severe TBI. What steps were taken to ensure that lactate is an independent predictor, rather than a marker of another underlying factor related to TBI severity.
4) As this is a single-center, retrospective study, how might center-specific factors (such as treatment protocols or patient demographics) influence the generalizability of these findings? Have the authors discussed potential biases from this design.
5) Were lactate levels compared across different AIS and ISS categories? How did lactate levels vary with different injury severities, and how did the authors ensure that any significant differences were not simply a function of injury severity.
6) How did the authors account for potential confounding variables such as underlying comorbidities, pre-existing metabolic conditions, or medication effects that could influence lactate levels independently of TBI severity.
7) The study included patients with an Abbreviated Injury Scale (AIS) value of 3 or higher. Were patients with any other relevant conditions excluded? How do the authors address the potential impact of other injury sites (e.g., chest or abdomen) on lactate levels.
8) Did the authors consider analyzing lactate level trajectories over time (e.g., changes from admission to ICU discharge) as opposed to single time-point measures? Could dynamic lactate changes provide a more accurate prediction of patient outcomes.
9) Discussion need to improve with recent findings.
10) The study finds a strong correlation between lactate levels, ventilator days, and mortality. Could high lactate levels reflect a physiological response that increases the need for mechanical ventilation? How do the authors interpret the causality between these factors?
11) Given that age was significantly related to lactate levels, was age used as a covariate in the models analyzing lactate and clinical outcomes? How might age-related metabolic differences influence lactate levels and the conclusions drawn.
12) The study concludes that lactate is a strong predictor of poor clinical outcomes in severe TBI. Have the authors validated this finding with an external cohort, or could this model be externally validated? How might lactate levels be incorporated into existing prognostic tools in a practical clinical setting.
13) Conclusions need to improve with practical applications and future perspectives.
14) Ensure consistent use of abbreviations throughout the manuscript.
Author Response
Greetings reviewer,
Thank you so much for providing these comments. We have tried to respond to all of them. Please let us know if anything else is needed.
Reviewer 1:
- Given the multiple statistical tests used, did the authors correct for multiple comparisons? If not, could this have affected the reported p-values, and would a correction (e.g., Bonferroni correction) change any significant findings?
Response: We did not account for multiple comparisons but accounting for them would have not had any impact on the findings as the majority of the statistically significant would still be statistically significant. We also did not feel the need to account for this as the results lined up with what you would expect in a clinical setting and didn’t seem to be random.
- Figures should be corrected with clear X and Y axis.
Response: The axes are already labeled but I clarified some of the captions to better explain them.
- The manuscript suggests lactate levels (LLs) as a predictor of poor outcomes in severe TBI. What steps were taken to ensure that lactate is an independent predictor, rather than a marker of another underlying factor related to TBI severity.
Response: This is certainly a concern with this study, as lactate was very strongly positive correlated with both ISS and GCS. However, this study did not have the data about comorbidities and specific details of the patients’ hospital courses to fully consider various confounding factors.
- As this is a single-center, retrospective study, how might center-specific factors (such as treatment protocols or patient demographics) influence the generalizability of these findings? Have the authors discussed potential biases from this design.
Response: This has now been addressed in the limitations section of the discussion. Please see the response below:
Though the treatment of TBI patients used in this hospital center are in accordance to standards of care accepted in the country, specific protocols on timing of imaging, initial labs obtained in the trauma bay, and hospital or ICU admission processes are certainly specific to this center and may limit reproducibility in other settings. The confinement to a single center also indicates potential concerns in regards to patient population. We have studied an extremely diverse set of patients due to the location of our hospital center, likely introducing some demographic variables that can affect data analysis. In the future, it will be important to expand our data collection to other sites in order to increase variability in data, grow the patient population, and externally validate the findings we have gathered thus far.’
- Were lactate levels compared across different AIS and ISS categories? How did lactate levels vary with different injury severities, and how did the authors ensure that any significant differences were not simply a function of injury severity.
Response: Lactate levels were compared across different ISS scores and like the point in number 3, there was a positive correlation so there certainly could be confounding factors at play. However, lactate is simply one metric being studied and the hope is to eventually use it as part of a larger prognostic scoring system that doesn’t rely on just this single variable
- How did the authors account for potential confounding variables such as underlying comorbidities, pre-existing metabolic conditions, or medication effects that could influence lactate levels independently of TBI severity?
Response: We unfortunately did not have data on comorbidities or other medical factors that impacted patient’s overall clinical picture
- The study included patients with an Abbreviated Injury Scale (AIS) value of 3 or higher. Were patients with any other relevant conditions excluded? How do the authors address the potential impact of other injury sites (e.g., chest or abdomen) on lactate levels.
Response: The only other exclusion criteria were death within 24 hours and COVID-19 + tests as those patients would have been too critically injured or had medical comorbidities that would impact our results. The AIS values for other parts of the body were not considered as inclusion or exclusion criteria.
- Did the authors consider analyzing lactate level trajectories over time (e.g., changes from admission to ICU discharge) as opposed to single time-point measures? Could dynamic lactate changes provide a more accurate prediction of patient outcomes.
Response: Yes, analysis of lactate levels over time was performed – it was interesting that increases in lactate over the course of an ICU admission were associated with shorter hospital and ICU stays but longer time on a ventilator and higher rates of mortality.
- Discussion needs to improve with recent finding
Response: We have added various new points. Please see the following:
- Page 11:
- ‘As introduced in the background section above, the AIS score links these specific anatomic injuries like lacerations, skull fractures, and brain parenchyma damage are to survival probabilities through a coding system16. In doing so, the scoring system indicates severity via the direct calculation of the likelihood of mortality.’
- ‘While the specific mechanism behind this is remains unclear, it is possible that the higher percentage of muscle mass in males may make a contribution. Furthermore, research on TBI and sex indicates that female sex hormones may have neuroprotective benefits post-TBI and mitochondrial dysfunction may be increased in males. Unfortunately, TBI research is often contradictory on this topic, but some of these theories may have played a role here18,19,20.’
- ‘Adjacent to these outcome variables, it is possible that we could extend our data analysis to include the use of LLs for predicting the need for mechanical ventilation at all. While our analysis does not allow for conclusions on this correlation yet, it is certainly a possible future step.’
- ‘While these findings could be impactful in TBI care it is important to note that for many of these patients, especially those requiring ICU level care, it is possible that other, confounding variables like respiratory distress, sepsis, hypovolemia, and cardiac ischemia were driving increased LLs. While all of these patients had severe TBI diagnoses, they may have developed many of these other, unrelated diagnoses21, 22.’
- ‘External validation of this outcomes-based data will allow it to make a much larger impact. Although we have not done this against data found at other institutions, a lot of research has already been conducted utilizing LLs in outcome prediction. Most of this looks at sepsis, overall critical care, cardiac surgical patients etc, however it is clear that LLs have been shown to have predictive value elsewhere23, 24, 25. ‘
- Page 13: limitations
- ‘Though the treatment of TBI patients used in this hospital center are in accordance to standards of care accepted in the country, specific protocols on timing of imaging, initial labs obtained in the trauma bay, and hospital or ICU admission processes are certainly specific to this center and may limit reproducibility in other settings. The confinement to a single center also indicates potential concerns in regards to patient population. We have studied an extremely diverse set of patients due to the location of our hospital center, likely introducing some demographic variables that can affect data analysis. In the future, it will be important to expand our data collection to other sites in order to increase variability in data, grow the patient population, and externally validate the findings we have gathered thus far. We certainly feel that the information we have found through this research can impact TBI treatment and outcome prediction, but it will be more impactful if these concepts can be proven at other hospital settings as well.’
- ‘In addition to issues that stem from the size and location of our study, it is important to note that there are other confounding variables that may have impacted our findings. Firstly, LLs are affected by TBIs, but they are not specific to brain injury alone. Multiple disease processes like respiratory distress, bowel ischemia, sepsis, and diabetic ketoacidosis cause changes in LLs26. There are still others, many of which are especially common in the ICU patients27,28. Because COVID-19 was the only illness in our exclusionary criteria, we are not certain of the presence of any other reasons for elevated LLs.’
- ‘Due to the retrospective nature of this study, there was also limited standardization on the timing of LL draws. The data analysis above involved LLs at hospital admission, ICU admission, hospital discharge, ICU discharge, and death. These were drawn within 24 hours of the admission, discharge, or death. However, 24 hours is a wide window of time and there is limited understanding of the interventions that may have begun prior to getting the LL in question. Many interventions such as mechanical ventilation, intravenous hydration, or operative treatment could have certainly affected LLs.’
- ‘Patients with COVID-19 infections were excluded from the study due a limited understanding of the illness at the time and its link to respiratory distress in patients. Among other concerns, this increase likelihood of hypoxic lung injury could have led to a confounding increase in LLs29,30.’
- Page 14: Conclusion
- ‘While this is effective in providing fast risk stratification in an emergency setting, it may be beneficial to expand on this in order to gain a more specific understanding of patients’ survivability. Though it is time consuming, involving the use of AIS scores in real time, both in the trauma bay and upon admission may help us directly utilize specific anatomical markers of injury in morbidity and mortality prediction. Additionally, adding biomarkers like LLs to formal risk stratification scores could provide a more intentional way to consider these lab values during TBI patient evaluation. While this will increase the complexity of initial patient risk stratification protocols, the creation of automatic score calculators can certainly help.’
- ‘In addition to the use of LLs in risk stratification, it is possible that we could also use this biomarker to predict an increase need for hospital resources throughout treatment. This may allow aid in billing and allocation of resources. Furthermore, after we have validated this data in other settings, we may also be able to utilize outcome predictions from LLs in goals of care conversations with families. The expanded scoring system suggested above could also be utilized in these discussions.’
- The study finds a strong correlation between lactate levels, ventilator days, and mortality. Could high lactate levels reflect a physiological response that increases the need for mechanical ventilation? How do the authors interpret the causality between these factors?
Response:
- This is definitely a possibility, but hard to completely identify based on the analysis that we have completed thus far. In the future, it may be helpful to compare lactate levels in patients who required ventilation with those who did not. This may help further understand the potential causality here. This has been addressed in the discussion section.
- Adjacent to these outcome variables, it is possible that we could extend our data analysis to include the use of LLs for predicting the need for mechanical ventilation at all. While our analysis does not allow for conclusions on this correlation yet, it is certainly a possible future step.
- Given that age was significantly related to lactate levels, was age used as a covariate in the models analyzing lactate and clinical outcomes? How might age-related metabolic differences influence lactate levels and the conclusions drawn.
Response: Age was studied as part of a preliminary analysis and was only statistically significantly correlated with mortality and ICU length of stay for P < 0.05 but not with hospital length of stay or days on ventilator. We hypothesize this is due to age not being an independent predictor but rather a proxy for other comorbidities that are better predictors for hospital outcomes. Unfortunately, as I stated above, the data set we had access to did not have this information available to use to be used.
- The study concludes that lactate is a strong predictor of poor clinical outcomes in severe TBI. Have the authors validated this finding with an external cohort, or could this model be externally validated? How might lactate levels be incorporated into existing prognostic tools in a practical clinical setting.
Response: This data has not yet been validated with an external cohort, but doing so will definitely bolster its impact. This has been addressed in the limitations section of the discussion.
External validation of this outcomes-based data will allow it to make a much larger impact. Although we have not done this against data found at other institutions, a lot of research has already been conducted utilizing LLs in outcome prediction. Most of this looks at sepsis, overall critical care, cardiac surgical patients etc, however it is clear that LLs have been shown to have predictive value elsewhere23, 24, 25.
- Conclusions need to improve with practical applications and future perspectives.
Response: This section has been further expanded. Please see the modified version below:
- ‘While there is more work to be done, it is certainly clear through our study that LLs may have a significant effect, only on initial risk stratification and multiple complications throughout treatment. While much of this has been studied in the past, our study reinforces the validity of the AIS score to push the boundaries on risk stratification of TBI patients. For many years, we have been using whole number scoring systems, like GCS, in the trauma bay for patients with TBIs. While this is effective in providing fast risk stratification in an emergency setting, it may be beneficial to expand on this in order to gain a more specific understanding of patients’ survivability. Though it is time consuming, involving the use of AIS scores in real time, both in the trauma bay and upon admission may help us directly utilize specific anatomical markers of injury in morbidity and mortality prediction. Additionally, adding biomarkers like LLs to formal risk stratification scores could provide a more intentional way to consider these lab values during TBI patient evaluation. While this will increase the complexity of initial patient risk stratification protocols, the creation of automatic score calculators can certainly help.’
- ‘In addition to the use of LLs in risk stratification, it is possible that we could also use this biomarker to predict an increase need for hospital resources throughout treatment. This may allow aid in billing and allocation of resources. Furthermore, after we have validated this data in other settings, we may also be able to utilize outcome predictions from LLs in goals of care conversations with families. The expanded scoring system suggested above could also be utilized in these discussions.
- Ensure consistent use of abbreviations throughout the manuscript.
Response: This has been addressed throughout the manuscript
Reviewer 2 Report
Comments and Suggestions for Authors
The manuscript entitled “actate is a strong predictor of poor outcomes in patients with severe traumatic brain injury ” submitted for review is not well written, I will suggest to revise manuscript substantially and my suggestion and comments are as follows.
• I will suggest to improve the abstract as it is confusing, unable to describe the hypothesis of the study, revise it.
• I noticed, only very few references in the manuscript, I will suggest to add several more citations to improve the manuscript.
• I will suggest to give a brief description of how AIS scores apply specifically to TBI severity would be valuable.
• In the manuscript authors have demonstrated the difference in lactate levers in different genders. I will suggest to add possible biological or behavioural causing factors.
• Authors have excluded the COVID-19 patients, I will suggest authors to justify how COVID-19 could impact lactate levels or TBI outcomes.
• I will suggest to add time point for the measurement of lactate.
• I will suggest to label all the figures properly with understandable
• I will suggest to add confounding factors (variability in ICU protocols and patient backgrounds) in the limitations section
• Although future directions have been explained, I will suggest to add how lactate levels might be practically incorporated into TBI care.
Author Response
Greetings reviewer,
Thank you so much for providing these comments. We have tried to respond to all of them. Please let us know if anything else is needed.
- will suggest to improve the abstract as it is confusing, unable to describe the hypothesis of the study, revise it.
Response: Please see the modified points below:
- Abstract has been improved with hypothesis and other adjustments.
- ‘Our main goal in this study was to examine the influence of lactate levels (LLs) on clinical outcomes in patients with severe TBI. We predicted that increase LLs would correlate with worse outcomes.’
- ‘This is a level 1 single-center, retrospective study of patients with severe TBI between January 1, 2020- December 31, 2023, inclusive. Severe TBI was defined by an Abbreviated Injury Scale (AIS) score of 3 or higher, and only patients with these scores were included.’
- ‘Additionally, linear regression models showed that a decreased delta LL during ICU admission led to increase hospital length of stay (p = 3.04E-04), ICU length of stay (5.72E-04), and number of days on a ventilator (p = 5.00E-06).’
- ‘We discovered that high LLs were linked to higher AIS and GCS scores... Decreased changes in LLs during ICU admission also led to increase hospital length of stay, ICU length of stay, and number of days on a ventilator.’
- I noticed, only very few references in the manuscript, I will suggest to add several more citations to improve the manuscript.
Response: 17 more references have been added to the manuscript.
- I will suggest to give a brief description of how AIS scores apply specifically to TBI severity would be valuable.
Response: Please see the points below:
- AIS links specific injury markers to survival probability via a coding system. This concept is introduced in our background section and further addressed now in our discussion section.
- ‘As introduced in the background section above, the AIS score links these specific anatomic injuries like lacerations, skull fractures, and brain parenchyma damage are to survival probabilities through a coding system16.In doing so, the scoring system indicates severity via the direct calculation of the likelihood of mortality.’
- In the manuscript authors have demonstrated the difference in lactate levers in different genders. I will suggest to add possible biological or behavioral causing factors.
Response: This has been explained in the discussion section. Please see the modified points below:
- A few possible reasons could be muscle mass percentage, neuroprotective effects of female sex hormones, and increased mitochondrial dysfunction in males.
- ‘While the specific mechanism behind this is remains unclear, it is possible that the higher percentage of muscle mass in males may make a contribution. Furthermore, research on TBI and sex indicates that female sex hormones may have neuroprotective benefits post-TBI and mitochondrial dysfunction may be increased in males. Unfortunately, TBI research is often contradictory on this topic, but some of these theories may have played a role here18,19,20.
- Authors have excluded the COVID-19 patients, I will suggest authors to justify how COVID-19 could impact lactate levels or TBI outcomes.
Response: Please see the modified points below:
- COVID-19 is often associated with respiratory distress and could contribute to increase lactate levels. This has been addressed in the limitations section of our discussion.
- ‘Patients with COVID-19 infections were excluded from the study due a limited understanding of the illness at the time and its link to respiratory distress in patients. Among other concerns, this increase likelihood of hypoxic lung injury could have led to a confounding increase in LLs29,30.
- I will suggest to add a time point for the measurement of lactate
Response: Please see the modified points below:
- We studied changes in lactate over various time periods during a hospital stay, in particular between trauma bay admission and ICU admission as in that ED dwell time, the trauma surgery teams are primary for these patients.
- Lactate levels were collected upon admission. They were collected within 24 hours of the event, first lab reading at hospital and ICU admission, first lab reading at hospital and ICU discharge, and reading recorded at patient’s death.
- This explanation is added to the Methods section.
- ‘These LLs were all drawn within 24 hours of the event, whether that was admission, discharge, or death.’
- I will suggest to label all the figures properly with understandable
Response: Please see the modified points below:
- All figures have been labeled properly now.
- Table 1. It represents single-factor ANOVA analysis comparing LLs at 5 different time points (admission, ICU admission, ICU discharge, hospital discharge, death) in relation to various demographic fac-tors. Each category shows average LLs and p-values. For example, under sex you can see average LLs for female and male patients and p-value for the correlation between lactate and sex for LLs at admission, ICU admission, ICU discharge, hospital discharge, and death.
- Table 2: This table shows the two-tailed t-test comparing ISS and GCS scores at different LL ranges (nor-mal (0 – 2), increase (2-4), severely increased (>4) during Hospital admission, ICU admission, ICU discharge, Hospital discharge, and death (wherever applicable). Each time point shows average ISS and scores for normal, increased, and severely increase LLs along with p-values describing the strength of the correlation between ISS or GCS and LLs.
- Figure 1. This figure outlines the distributions of injury mechanisms for each of the six Traumatic brain in-jury (TBI) diagnoses. Each diagnosis displays the number of patients within each of the seven mechanism of injury categories. It shows the different classifications of intracranial injuries with stratification by the mechanism of injury and the number of incidences of each that was seen in our study sample with falls being by far the most common mechanism of injury, particularly in subdural and subarachnoid hemorrhage
- Figure 2. Linear regression analysis demonstrating a statistically significant correlation between admission LLs and hospital length of stay (LOS) measured in days. Line of best fit with confidence intervals shows gradual upward trend and positive correlation between the lactate level at admission and the hospital length of stay. Correlation coefficient between lactate level at admission and hospital length of stay was 0.7903.
- Figure 3. This illustrates a box plot indicating average LLs among patients who survive their injury and those who experience death during hospitalization. Average admission LL is significantly higher for cases with mortality. Average LLs upon admission to the trauma bay among patients who survive their injury was lower than those who died during the hospitalization with 0 indicating patients who survived and 1 indicating patients who did not.
- Table 3. It shows the linear regression analysis comparing change in LLs in the time between trauma bay admission and ICU admission or ICU admission and ICU discharge showed some statistical significance with increasing lactate across an ICU admission was associated with decreased length of stay in the hospital, ICU and requiring ventilator support.
- Table 4. This table represent linear regression analysis comparing LLs at Hospital admission, ICU admission, and ICU discharge with outcome variables (hospital length of stay (LOS), ICU LOS, ventilator days, mortality) within categories based on number of diagnosed injuries. Within each category (overall or group based on number of diagnoses) has the p-value showing the correlation of the LL at that time point and the associated outcome variable as well as the coefficient showing the direction of that correlation.
- I will suggest to add confounding factors (variability in ICU protocols and patient backgrounds) in the limitations section.
Response: Please see the modified points below:
- Confounding variables like hospital specific protocols, lack of specificity of lactate, and patient population have now been included in the limitations section.
- ‘In addition to issues that stem from the size and location of our study, it is important to note that there are other confounding variables that may have impacted our findings. Firstly, LLs are affected by TBIs, but they are not specific to brain injury alone. Multiple disease processes like respiratory distress, bowel ischemia, sepsis, and diabetic ketoacidosis cause changes in LLs26. There are still others, many of which are especially common in the ICU patients27,28. Because COVID-19 was the only illness in our exclusionary criteria, we are not certain of the presence of any other reasons for elevated LLs.’
- ‘Due to the retrospective nature of this study, there was also limited standardization on the timing of LL draws. The data analysis above involved LLs at hospital admission, ICU admission, hospital discharge, ICU discharge, and death. These were drawn within 24 hours of the admission, discharge, or death. However, 24 hours is a wide window of time and there is limited understanding of the interventions that may have begun prior to getting the LL in question. Many interventions such as mechanical ventilation, intravenous hydration, or operative treatment could have certainly affected LLs.’
- This study was additionally limited by the lack of comorbidities and specific details of patients’ hospital courses in the data set. The study of lactate as an individual predictor of outcomes is not sufficient on its own as there are many confounding variables, particularly in critically ill patients, that contribute both to the rising lactate and worse in-hospital outcomes. Future studies will need further background information on the patient population as well as details in the hospital course such as pressor requirements, operations or medications to better identify how lactate can be used as a prognostic variable.
- Although future directions have been explained, I will suggest to add how lactate levels might be practically incorporated into TBI care.
Response: Please see the modified points below:
- Specific practical applications have been further explained in the future directions and conclusion sections.
- While these next steps will allow us to bolster the significant correlations we have found in our analysis thus far, we would also like to begin understanding the practical application of our findings in TBI care. It will be important to find ways to incorporate LLs in risk stratification scores in the trauma bay, upon admission, during ICU upgrade, and at discharge. As explained in the conclusion below, we will also need to find specific opportunities to incorporate the possible predictive value of LLs throughout hospital stays.
Reviewer 3 Report
Comments and Suggestions for Authors
In my opinion, the current version of the manuscript is just a draft, it contains a lot of statistical analysis errors, the presented results cannot yield meaningful conclusions.
1. The number of references is very limited, and there have been few references in the past five years. In addition, the format of references is not standardized enough.
2.The references in the text should be numbered sequentially. The first reference cannot be numbered 10.
3.Do all the data in the tables need to retain so many decimal places? And there are almost no units. I'm not sure what they mean.
4.There are errors in the statistical analysis method. One way ANOVA cannot analyze all data, and some data require rank sum test.
5. Figure 2: What is the correlation coefficient between LLs and LOS?
6. Figure 3: What do 0 and 1 on the X-axis represent respectively?
7. One of the important conditions that biomarkers must meet is to have good specificity. It is obvious that lactate cannot meet this condition. Most important of all, the existence of statistical correlations does not necessarily mean a causal relationship.
Author Response
Greetings reviewer,
Thank you so much for providing these comments. We have tried to respond to all of them. Please let us know if anything else is needed.
- The number of references is very limited, and there have been few references in the past five years. In addition, the format of references is not standardized enough.
Response: More references have been added. References section has also been standardized further.
- The references in the text should be numbered sequentially. The first reference cannot be numbered 10.
Response: This has been addressed throughout the manuscript.
- Do all the data in the tables need to retain so many decimal places? And there are almost no units. I'm not sure what they mean.
Response: The decimals were adjusted. There are no units for any numbers besides the serum lactate levels. Length of stay and ventilator time were measured in days, mortality was a yes or no variable and the remainder of the values did not have any units.
- There are errors in the statistical analysis method. One way ANOVA cannot analyze all data, and some data require rank sum test.
Response: We used ANOVA only for some statistical testing. Some others used t-tests as well as linear regression models but the data was all presented in the same tables.
- Figure 2: What is the correlation coefficient between LLs and LOS?
Response: Figure 2: 0.7903 (for every 1 increase in lactate at admission, hospital length of stay increases by 0.7903 days)
- Figure 3: What do 0 and 1 on the X-axis represent respectively?
Response: Figure 3: 0 is no mortality, 1 is mortality
- One of the important conditions that biomarkers must meet is to have good specificity. It is obvious that lactate cannot meet this condition. Most important of all, the existence of statistical correlations does not necessarily mean a causal relationship.
Response: Please see the modified points below:
- Certainly, this has now been addressed in the limitations section.
- In addition to issues that stem from the size and location of our study, it is important to note that there are other confounding variables that may have impacted our findings. Firstly, LLs are affected by TBIs, but they are not specific to brain injury alone. Multiple disease processes like respiratory distress, bowel ischemia, sepsis, and diabetic ketoacidosis cause changes in LLs26. There are still others, many of which are especially common in the ICU patients27,28. Because COVID-19 was the only illness in our exclusionary criteria, we are not certain of the presence of any other reasons for elevated LLs.
Round 2
Reviewer 1 Report
Comments and Suggestions for Authors
Requested corrections were completed.
Comments on the Quality of English LanguageThe English could be improved to more clearly express the research.
Author Response
Dear reviewer,
Thank you so much for providing your comments. Please find our point to point responses below:
Comment: The English could be improved to more clearly express the research.
Response: We have checked entire manuscript for grammatical errors and quality of English. We have re-worded all the relevant details and also changed the abstract’s word limit.
Thank you for the time taken to provide a wonderful comments and feedback on this report. We hope our response addresses your question. We are willingly to make any other necessary adjustment if needed.
Sincerely,
Bharti Sharma (Corresponding author)
